# Velocity-Selective Arterial Spin Labeling Perfusion in Monitoring High Grade Gliomas Following Therapy: Clinical Feasibility at 1.5T and Comparison with Dynamic Susceptibility Contrast Perfusion

**DOI:** 10.3390/brainsci14020126

**Published:** 2024-01-25

**Authors:** Sebastian Lambrecht, Dapeng Liu, Omar Dzaye, David O. Kamson, Jonas Reis, Thomas Liebig, Matthias Holdhoff, Peter Van Zijl, Qin Qin, Doris D. M. Lin

**Affiliations:** 1Russell H. Morgan Department of Radiology and Radiological Science, The Johns Hopkins University School of Medicine, Baltimore, MD 21287, USA; sebastian.lambrecht@mri.tum.de (S.L.); dliu48@jhmi.edu (D.L.); vanzijl@kennedykrieger.org (P.V.Z.); qqin1@jhmi.edu (Q.Q.); 2Institute of Neuroradiology, University Hospital LMU Munich, 81377 Munich, Germany; jonas.reis@med.uni-muenchen.de (J.R.); thomas.liebig@med.uni-muenchen.de (T.L.); 3F. M. Kirby Research Center for Functional Brain Imaging, Kennedy Krieger Institute, Baltimore, MD 21205, USA; 4Department of Oncology, The Johns Hopkins University School of Medicine, Baltimore, MD 21287, USA; odzaye@jhmi.edu (O.D.); dkamson1@jhmi.edu (D.O.K.); mholdho1@jhmi.edu (M.H.)

**Keywords:** velocity-selective arterial spin labeling (VSASL), arterial spin labeling (ASL), dynamic susceptibility contrast (DSC), perfusion, glioma, glioblastoma

## Abstract

MR perfusion imaging is important in the clinical evaluation of primary brain tumors, particularly in differentiating between true progression and treatment-induced change. The utility of velocity-selective ASL (VSASL) compared to the more commonly utilized DSC perfusion technique was assessed in routine clinical surveillance MR exams of 28 patients with high-grade gliomas at 1.5T. Using RANO criteria, patients were assigned to two groups, one with detectable residual/recurrent tumor (“RT”, n = 9), and the other with no detectable residual/recurrent tumor (“NRT”, n = 19). An ROI was drawn to encompass the largest dimension of the lesion with measures normalized against normal gray matter to yield rCBF and tSNR from VSASL, as well as rCBF and leakage-corrected relative CBV (lc-rCBV) from DSC. VSASL (rCBF and tSNR) and DSC (rCBF and lc-rCBV) metrics were significantly higher in the RT group than the NRT group allowing adequate discrimination (*p* < 0.05, Mann–Whitney test). Lin’s concordance analyses showed moderate to excellent concordance between the two methods, with a stronger, moderate correlation between VSASL rCBF and DSC lc-rCBV (r = 0.57, *p* = 0.002; Pearson’s correlation). These results suggest that VSASL is clinically feasible at 1.5T and has the potential to offer a noninvasive alternative to DSC perfusion in monitoring high-grade gliomas following therapy.

## 1. Introduction

MR perfusion imaging, particularly using dynamic susceptibility contrast (DSC) perfusion-weighted imaging (PWI) is an important component of the clinical evaluation of primary brain tumors. Elevated relative cerebral blood volume (rCBV) derived from DSC perfusion has been shown to correlate well with higher histological grades of tumors and with overall survival [1,2,3,4]. Furthermore, DSC perfusion adds clinical diagnostic confidence in differentiating between true progression and treatment-induced change that is difficult to achieve using conventional T1 post-Gadolinium MR images [5]. DSC perfusion is easy and fast to perform, particularly with the gradient-echo echo-planar imaging (EPI) technique now commonly available in clinical settings, and offers high signal-to-noise ratio (SNR) compared to the spin echo technique [6]. One pitfall of DSC perfusion is the effect of contrast leakage on quantification, which may occur in high-grade neoplasms due to abnormal vessels from neoangiogenesis and breakdown of the blood–brain barrier (BBB) [7,8]. This is more pronounced for rCBV measurements than relative cerebral blood flow (rCBF) [9] and can be reduced by leakage correction algorithms [1]. Additional pitfalls include the influence of large vessels and neovascularization [10] or susceptibility artifacts [11]. These limitations are particularly salient in the post-treatment evaluation, as the perfusion data are further affected by treatment related inflammatory change and hemorrhage.

The noncontrast arterial spin labeling (ASL) perfusion methods have lagged as an alternative for widespread clinical application because of a relatively low SNR (particularly at lower field strength, 1.5T), increased time requirement and concomitant sensitivity to motion-based image degradation [12,13]. Furthermore, the technical demands for image acquisition such as optimal placement of a labeling plane that is needed for pseudo-continuous ASL (PCASL) and the limited availability of these sequences has hampered clinical implementation of these methods. However, ASL is attractive in the evaluation of brain tumors because of less prominent susceptibility artifacts [14] and the use of water as an endogenous diffusible tracer that can pass capillary membranes [13]. Additionally, it offers a more quantitative assessment of CBF that can be repeatedly measured. In the rare occasions that Gadolinium contrast agent is contraindicated or undesirable (e.g., pregnancy, renal failure, contrast reaction, or lack of intravenous access), ASL provides an alternative means for perfusion measures [15,16,17].

Velocity-Selective Arterial Spin Labeling (VSASL) is a relatively new ASL technique using velocity encoding rather than spatial encoding [18,19]. Its advantages of achieving high SNR with minimal sensitivity to slow flow artifacts have been demonstrated among patients with cerebrovascular diseases [20,21]. Technical challenges for the conventional VS saturation (VSS) based VSASL included sensitivity to B0/B1 field inhomogeneity [18,19] and eddy currents [22,23]. Poor perfusion sensitivity and small spatial coverage limited conventional VSASL from widespread utilization. The Fourier transform (FT) based VS inversion (VSI) pulse train was designed with paired refocusing pulses and phase cycling for improved immunity to B0/B1 field inhomogeneities and eddy currents, and was applied to CBF mapping showing 40% improvement of SNR comparing to VSS based VSASL [24]. Three-dimensional GRASE acquisition was subsequently combined with VSI labeling, which showed its superior perfusion sensitivity and image quality compared to PCASL [25], and excellent test–retest reliability to detect both between-subject and between-region normal variations [26]. A dynamic phase-cycling scheme for VSASL was proposed to further ensure both velocity and spatial responses robust to field inhomogeneities [27]. Moreover, the effectiveness of FT-VS labeling has been demonstrated for mapping CBV [28,29,30,31], venous oxygenation [32], MR angiography [33,34,35,36,37], and with a commercial perfusion phantom [38]. Other groups have also confirmed better performance in detecting perfusion signal changes under brain stimulus or with respect to temporal SNR by FT-VSI ASL than by PCASL [39,40,41]. The first guideline paper on VSASL [42] acknowledged that “FT-VSI pulses yield greater labeling efficiency if B0 and B1 fields are reasonably homogeneous (e.g., in the head)”. More recently the use of VSASL has been shown to provide perfusion measures comparable to PCASL and DSC in newly diagnosed brain tumors, and was found to be able to distinguish between low- and high-grade tumors in a cohort of 44 patients with primary glioma [43].

Distinguishing recurrent tumor from pseudo-progression or radiation necrosis in post-treatment monitoring remains a challenging task but can be assisted by MR perfusion measurements, as tumor recurrence leads to neovascularization and therefore increased tissue perfusion (as indicated by blood volume and blood flow) [44], while radiation damage is characterized by effects such as fibrinoid changes in blood vessels and coagulative necrosis resulting in diminished tissue perfusion [45]. For this purpose, several studies have been conducted to determine the cutoff thresholds of DSC-PWI-derived parameters for discriminating tumor recurrence from treatment effect [46,47]. The utility of ASL and especially VSASL in monitoring disease following treatment of glioma is, however, thus far much less explored [48,49]. The purpose of this study was to evaluate the feasibility of incorporating FT-VSI based VSASL into the routine clinical exam of gliomas at 1.5T (most widely available field strength), and compare the information derived from FT-VSI based VSASL with DSC perfusion.

## 2. Materials and Methods

### 2.1. Study Design and Patient Population

This prospective study was approved by our Institutional Review Board and written informed consent was obtained from each patient. Twenty-nine consecutive patients with high-grade glioma presenting for routine follow-up MRIs at our Outpatient Cancer Center between June and August 2019, were recruited. One patient was excluded due to a failed Gadolinium injection rendering DSC perfusion nondiagnostic, yielding a final dataset of 28 patients for analysis. Motion artifacts were not considered exclusion criteria as we aimed to investigate the clinical feasibility of both FT-VSI based VSASL and DSC-PWI. Three patients had subsequent surgery within 4 weeks of MRI for histopathological confirmation. Presence or absence of detectable residual or recurrent tumor was determined by clinical exams including overall clinical status, steroids use, as well as radiological assessment, following standard RANO criteria [50], based on the index exam and subsequent follow-up MRIs.

### 2.2. Imaging Protocol

All MRI scans were performed at 1.5 Tesla Siemens Magnetom Aera system (Siemens Healthineers, Erlangen, Germany) with a 20-channel head coil. The clinical brain glioma protocol consists of T1, T2, FLAIR, DWI, SWI, DSC perfusion, and postcontrast axial and coronal T1 weighted spin echo sequences. This protocol utilized the accelerated anatomic sequences and the entire study required an average of 17 min scan time.

Prior to the injection of contrast, a FT-VSI based VSASL perfusion map was acquired using protocol similar to a previous study conducted at 3T [25]. The cutoff velocity for labeling was 2.8 cm/s along the foot–head direction. The postsaturation delay (interval between slab-selective saturation module and label/control module) was set as 2.0 s. The postlabeling delay was chosen as 1.0 s. Parameters for the 3D GRASE readout were: excitation flip angle = 90°; refocusing flip angle = 120°; acquisition = 3.4 × 3.4 × 4 mm^3^, reconstructed to 1.7 × 1.7 × 4 mm^3^; FOV = 220 × 220 × 96 mm^2^; bandwidth = 2894 Hz per pixel; TE/TR = 16.5/3300 ms; EPI factor = 23; TSE factor = 10; six segments acquired for each dynamic; a total of six dynamics (label and control pairs); scan time 4 min 11 s. A proton-density-weighted M0 image was also acquired by disabling labeling and setting TR to 4.0 s (total 32 s).

DSC perfusion images were recorded during bolus injection of 0.1 mmol/kg of Gadavist (Gadobutrol, Bayer Healthcare, Berlin, Germany) at 5 mL/s, followed by 30 mL of normal saline flush at the same rate, by using single-shot gradient-echo EPI (TR/TE = 2000/30 ms; 60 dynamics; flip angle = 90°; matrix = 120 × 96; slice thickness = 4 mm) with whole brain coverage. Scan time was 2 min.

### 2.3. Data Analysis

The raw ASL data were postprocessed using MATLAB R2017a (The MathWorks Inc., Natick, MA, USA) to generate rCBF and temporal SNR (tSNR) maps. Rigid motion correction was performed with the object function of normalized mutual information using SPM12 (The Wellcome Centre for Human Neuroimaging, UCL Queen Square Institute of Neurology, London, UK). Each label and control image was registered to its M0 image, respectively. DSC perfusion raw data were processed using Olea Sphere v.3.0.20 (Olea Medical, La Ciotat, France) to produce semiquantitative (relative) parametric maps of rCBF, and leakage-corrected relative CBV (lc-rCBV), including automated arterial input function detection and oscillatory singular value decomposition deconvolution based on the assumption that the input data describe a well-defined signal response (first-pass) following contrast bolus. The rCBF and rCBV values were estimated, and their units are arbitrary. Absolute quantification was not feasible because the proportionality constants required in the conversion of signals into concentration–time curves were unknown. All DSC perfusion maps and ASL data (after motion-correction as previously described) were resliced with SPM12 to match the resolution and position of anatomic images, so that ROIs can be shared between all of them to a good approximation.

An ROI was drawn for each case on one single axial slice covering the largest dimension of tumor/lesion using ImageJ 1.52a to encompass areas of postcontrast T1 and FLAIR hyperintensity, while avoiding large vessels (see Figure 1a). When no detectable enhancing lesion was present, an ROI was drawn in the treatment field containing FLAIR hyperintensity adjacent to the resection cavity. As a reference, unaffected, normal-appearing brain tissue was selected either from the entire contralateral hemisphere, or from unaffected brain on a different slice, excluding any portion with visible hyperperfusion related to large vessels. Within these large reference ROIs, the grey matter was segmented using specific thresholds generated by MATLAB and confirmed by visual inspection.

For VSASL, the VSASL-derived perfusion-weighed signal (PWS) maps were divided by the mean PWS of the segmented normal grey matter to yield rCBF maps. PWS was proportional to the true CBF; however, absolute quantification was not feasible at this time since the labeling efficiency of VSASL at 1.5T remained unknown. tSNR was derived from the quotient of the mean and the standard deviation of the signal from the transient time series. To be consistent with ASL, DSC rCBF and lc-rCBV values obtained from each ROI were also divided by the mean values of the segmented normal grey matter.

The final mean lesion metrics were calculated from automatically selecting 10 pixels with the highest perfusion values within the lesion ROI and normalized against the mean perfusion values of the segmented grey matter, yielding rCBF-ratio and tSNR-ratio for VSASL; rCBF-ratio and lc-rCBV-ratio for DSC-PWI.

### 2.4. Statistical Analysis

A Mann–Whitney test was performed for each perfusion method to compare measured values between detectable residual/recurrent tumor (RT) and no-detectable residual/recurrent tumor (NRT) cases. Correlation of VSASL rCBF-ratio and DSC rCBF-ratio was tested using Pearson’s correlation. Furthermore, a linear regression was performed on the correlation data. A spatial correlation was performed for the patient group with residual/recurrent tumor, comparing the two perfusion methods by calculating Pearson’s r for each voxel within the ROIs. Significance level was set at α = 0.05. All statistical analyses were performed using GraphPad Prism 8.4.3.

## 3. Results

A total of 28 patients in this study included 12 females and 16 males, with mean age of 55.2 years (range 22–79 years), all with previously treated high-grade gliomas: 16 Grade IV, 12 Grade III according to WHO 2016 classification [51]. Twenty-six out of the twenty-eight patients underwent surgery, chemoradiation, and adjuvant chemotherapy with Temozolomide with a detailed clinical profile shown in Appendix A. Using a combination of imaging and clinical assessment and, if available, histopathology (in three patients), the patients were placed into two groups: “RT” (n = 9, 100% IDH wildtype, 78% Grade IV) and “NRT” (n = 19, 13 with known IDH status: 69% IDH wildtype, 47% Grade IV). The mean patient age in the RT group was 52.4 years, not significantly different from the NRT group of 56.5 years. Time since initial diagnosis averaged 894.8 days for the tumor group, significantly shorter than the 2796.8 days in the NRT group (*p* = 0.028, 2-tailed *t*-test).

Figure 1 shows two examples of DSC and VSASL-derived parametric maps at the sites of viable tumor with elevated signal. Figure 1a illustrates a case of newly diagnosed recurrent glioblastoma (IDH1 wild-type, MGMT methylated) on a 6-month surveillance MRI in a 54-year-old man who had undergone gross total resection 6.5 years ago, and completed concurrent radiation and temozolomide followed by six cycles of adjuvant temozolomide. An irregular enhancing infiltrative mass in the left parietal lobe near the resection cavity was characterized by elevated perfusion on rCBF and lc-rCBV derived from DSC, as well as rCBF and tSNR from VSASL. Subsequent exams showed progressive enlargement of the enhancing mass, confirming progressive disease.

Figure 1b shows a surveillance scan of a 70-year-old man with a history of right temporal glioblastoma at 13 months following surgery. A new enhancing mass in the contralateral left anterior temporal lobe was marked by elevated perfusion that was clearly present on all maps and confirmed by surgery 12 days later to represent active, highly cellular glioblastoma with pseudopalisading necrosis. At the site of previous glioblastoma resection, treatment change in the anterior right temporal lobe was characterized by diminished lc-rCBV and rCBF on both DSC and VSASL (Figure 1b). On a more superior image slice Figure 1c, however, there was increased heterogeneous contrast enhancement involving right frontotemporal region, showing moderately elevated perfusion on DSC (lc-rCBV and rCBF) and VSASL rCBF and tSNR. Surgical pathology also confirmed active glioblastoma and extensive necrosis.

Figure 2 shows that perfusion measures derived from both VSASL (rCBF-ratio and tSNR-ratio) and DSC (rCBF-ratio and lc-rCBV-ratio) are useful in discriminating RT from NRT cases. Based on VSASL, the median rCBF-ratio value of 1.17 (interquartile range, IQR 0.81–1.52) in NRT cases was significantly lower than 2.09 (1.52–3.09) in RT (*p* = 0.0016, Mann–Whitney test, Figure 2a). tSNR-ratio had a median value of 1.36 (0.87–2.10) in NRT compared to 2.50 (1.65–5.31) in RT (*p* = 0.0013, Figure 2b). Based on DSC, the median rCBF-ratio value of 1.39 (1.16–1.56) in NRT cases was significantly lower than 2.11 (1.69–2.90) in RT (*p* = 0.0013, Figure 2c). Similarly, the median lc-rCBV-ratio value of 1.39 (1.12–1.68) in NRT was significantly lower than 2.19 (1.47–2.37) in RT (*p* = 0.022, Figure 2d).

Lin’s concordance plot in Figure 3 shows moderate concordance between VSASL rCBF-ratio and DSC rCBF-ratio for the tumor recurrence cases (r = 0.42, *p* = 0.026; Pearson’s correlation) and stronger, moderate concordance between VSASL rCBF-ratio and DSC lc-rCBV-ratio (r = 0.57, *p* = 0.002). Furthermore, DSC rCBF-ratio and VSASL tSNR-ratio were significantly correlated with r = 0.44 (*p* = 0.019); while DSC lc-rCBV-ratio and VSASL tSNR-ratio showed the highest correlation with r = 0.81 (*p* < 0.0001), shown in Figure 3d.

Scatter plots of spatial correlation between VSASL rCBF and DSC rCBF or DSC lc-rCBV (Figure 4) were generated from the pooled ROIs encompassing tumor/lesion. These plots show that VSASL rCBF and DSC rCBF or lc-rCBV have a significant (*p* < 0.0001), moderately high spatial correlation between the different parameters in depicting the area of elevated perfusion.

## 4. Discussion

The results demonstrate that VSASL provides relative perfusion measures comparable to DSC-PWI in the evaluation of treated primary gliomas. Various parameters including rCBF-ratio and tSNR-ratio derived from VSASL, and lc-rCBV-ratio and rCBF-ratio from DSC-PWI show moderate to excellent correlation and concordance between the two methodologies. For direct comparison, we have included rCBF-ratio measurements from both methods in this study, even though lc-rCBV from DSC-PWI is the primary parameter most frequently used for tumor vascularity reflecting neoangiogenesis, and as an image marker associated with aggressiveness and therefore grades of primary gliomas. Here we found that the leakage-corrected rCBV is colinear with rCBF and shows a positive correlation with VSASL rCBF. The stronger correlation between DSC lc-rCBV-ratio and VSASL rCBF-ratio is likely attributable to a more reliable estimate in the calculated DSC lc-rCBV compared to DSC rCBF; in DSC-PWI, CBF requires the deconvolution of the tissue curves by an arterial input function, which can be a source of error depending on the choice of arteries and perfusion model. In characterization of malignant tumor perfusion using DSC methods, there are added challenges because of BBB disruption leading to contrast extravasation, which is further confounded in treated gliomas following surgery and chemoradiation. Contrast extravasation affects both T1 and T2* in tissues and extravascular water, therefore altering the apparent ∆R2* after contrast bolus and potentially causing either under- or over-estimation of derived parameters, including CBV and CBF. While CBV is corrected by the Olea software (v.3.0.20) for contrast leakage (expressed as corrected rCBV using the Weisskoff method), CBF is not. DSC-derived CBF measurements may therefore be prone to estimation errors, particularly in tumor cases where there is BBB breakdown either from tumor neovascular proliferation or treatment effects, or a combination of both. The contrast extravasation-induced error is not an issue when ASL perfusion is used, hence VSASL-rCBF and leakage-corrected DSC-rCBV show a higher correlation; this also confirms the results from other tumor perfusion studies [52]. In some cases, VSASL-based CBF elevation in tumors was less conspicuous than on DSC perfusion; one potential explanation of this may be that low arterial blood velocity in the treated tumor resulted in lack of labeling, based on the cutoff velocity used in the current study (2.8 cm/s).

Consistent with the previous studies showing the utility of ASL and DSC in delineating and characterizing primary brain tumors, VSASL was capable of discriminating tumors from surgical bed and treatment changes by quantitative measures, in this case a ratio, as the labeling efficiency was unknown. In our cohort, the performance of VSASL was comparable to DSC and may offer some advantages, as DSC can be limited in some cases by the strong influence from large vessels, contrast leakage, and sensitivity to susceptibility artifacts. Susceptibility artifact may be an important concern in treated high-grade gliomas because of hemorrhage related to tumor or surgery.

Since ASL requires a longer acquisition time (5 min ASL vs. 2 min DSC) and requires subtraction of images with and without labeling, it is more prone to motion artifact. This may be mitigated in part by some of the approaches used in this study. First, retrospective motion correction using SPM was simple and robust, and was performed in a few minutes, although this did still require offline postprocessing. Second, scan time was reduced by limiting the volume of coverage to the site of tumor (i.e., regional ASL). In the future, scan time could also be reduced by decreasing the number of segments, although this would be at the expense of SNR. Moreover, future developments by speeding up acquisition would further augment the clinical feasibility and utility of ASL.

ASL in general has the disadvantage of a lower SNR compared to DSC technique; however, the current VSASL protocol has shown a good tSNR. A potential advantage of tSNR is that it may depict the more stably elevated perfusion signal in tissue, distinguishing it from vascular-related signal that is subject to greater temporal variation. Conceivably, tSNR also more specifically highlights perfusion into cellular tumor than treated tissues with predominantly inflammatory changes so that it correlates with leakage-corrected DSC-rCBV better than the other parameters (Figure 3d). The utility of tSNR in assessing the fraction of viable tumor vs. post-treatment tissue necrosis would be best evaluated in future prospective studies by correlating with histopathologic specimens from patients with previously treated glioblastoma undergoing repeated surgical resection. Recently, 3D single-shot VSASL was achieved using stack-of-spiral FLASH acquisition and compressed-sensing acceleration with high temporal resolution [53], which could further improve the tSNR analysis.

At present, 3T is widely used in clinical MRI scanning and has allowed improvements in SNR for both spatial- and velocity-encoded ASL techniques. There is an added advantage at 3T because of the longer blood T1 resulting in increased label lifetime. Higher field strength further provides advantages for accelerated scans while maintaining similar spatial resolution. Most of the technical developments for VSASL were, in fact, made on 3T. The currently used 3D GRASE acquisition combined with FT-VSI labeling has demonstrated SNR close to PCASL at 3T [25]. Clinical translation of VSASL at 3T was applied to characterize perfusion properties in newly diagnosed primary glioma [43], showing a more comparable image contrast to DSC than PCASL on visual inspection, as well as a higher correlation of derived tumor CBF measures between VSASL and DSC on quantitative analysis. While all three techniques allowed good discrimination between low- and high-grade tumors, the same study using receiver operating characteristic curves showed that VSASL had superior diagnostic sensitivity, specificity, and accuracy in tumor grading compared to PCASL.

Compared to PCASL, VSASL has the advantage of being less technically demanding (in terms of properly prescribing labeling pulses) and therefore less prone to acquisition errors. This was repeatedly demonstrated in our early phase of the study, as many PCASL acquisitions were nondiagnostic while VSASL provided useful information. VSASL is also more robust compared to spatially labelled ASL techniques (such as PCASL) at 1.5T as it theoretically can eliminate arterial transit delay between label and image acquisition across the imaging volume, thereby minimizing signal loss from T1 relaxation. ASL is an intrinsically low SNR perfusion technique, and spatially labeled ASL perfusion scans require a compromise between SNR and sufficiently long postlabeling delay that allows tagged blood to reach the capillary bed and exchange with tissues. PCASL is therefore limited in regions with low or delayed flow, and is dependent of blood T1 as well as age (with longer arterial arrival times in older individuals). In contrast, the penalty on SNR is less severe for VSASL because of its reduced sensitivity to arterial transit time.

Finally, we were able to perform reasonable quality VSASL on all 29 patients, while Gadolinium injection for the DSC-PWI failed for one patient. Although DSC-PWI is easy to perform, it can be considered ‘minimally invasive’ as it requires intravenous contrast administration. Technical failure or suboptimal acquisition, even though rare in experienced centers, may at times be inevitable caused by physiologic conditions such as patient’s poor intravenous access or poor cardiac output.

There are limitations in this study. First, a relatively small cohort of 28 patients were included. Second, there were unbalanced numbers in the two groups, with most patients being in the nontumor group. Third, since this cohort was drawn from patients undergoing routine follow-up exams after treatment, typically there was no subsequent surgery for definitive histopathologic confirmation; therefore, designation of the tumor vs. nontumor groups is primarily based on clinical means aided by RANO criteria. Nevertheless, this has been an accepted practice in a number of clinical trials. Fourth, the DSC protocols used here were as supplied by the manufacturer, and did not exactly match the currently recommended DSC protocols for glioma [54] at 1.5T; however, we believe that similar overall conclusions would be found when using the consensus-recommended DSC protocol. Even with a small cohort, the current study shows clinical performance of VSASL comparable to DSC-PWI, including its utility of supplementing conventional scans with tumor perfusion information in the treated bed, a task that is most challenging in neuro-oncology and yet critically important. To further evaluate the clinical performance of VSASL, a larger cohort will be needed.

## 5. Conclusions

In conclusion, VSASL is clinically feasible at 1.5T and may be a noninvasive alternative to DSC-PWI in monitoring disease in high-grade gliomas following therapy. Especially, VSASL-based tSNR showed a strong correlation with leakage-corrected rCBV from DSC. Use of ASL is particularly valuable when Gadolinium contrast is contraindicated or undesirable. Further research is needed to validate this perfusion method in a larger cohort for its robustness in distinguishing tumor from largely treatment effects, and to develop faster acquisition as well as streamlined postprocessing routines.

## Figures and Tables

**Figure 1 brainsci-14-00126-f001:**
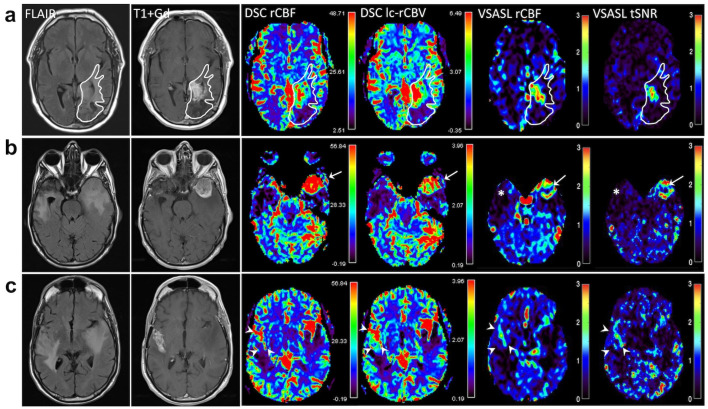
(**a**) Region of interest (ROI) encompassing the treated primary tumor site was manually drawn on one single axial slice based on anatomic images (FLAIR and T1 + Gd). The same ROI was transposed on perfusion parametric maps (rCBF and lc-rCBV from DSC, and rCBF and tSNR from VSASL) while avoiding large vessels. The mean lesions rCBF, rCBV, and tSNR were then calculated using a custom MATLAB code by automatically selecting 10 pixels with the highest perfusion values within the lesion ROI and normalized against the mean reference grey matter CBF, expressed as ratios, for quantitative analysis. Color scales for DSC rCBF and lc-rCBV are shown in arbitrary units; VSASL rCBF and tSNR are ratios and thus their color scales are dimensionless. The irregularly enhancing tumor in the left temporo-occipital region shows elevated signal on DSC and VSASAL maps. (**b**) An example of right temporal glioblastoma resection site showing encephalomalacia corresponding to diminished perfusion shown on DSC and VSASL (*). In the contralateral left anterior temporal lobe is a new circumscribed enhancing mass with surrounding edema, showing elevated perfusion on all perfusion maps: DSC rCBF and lc-rCBV, and VSASL rCBF and tSNR (arrows). (**c**) More cephalad to the right temporal resection in the same patient as in (**b**) is heterogeneous contrast enhancement extending to the right Sylvian fissure involving right frontotemporal region (arrowheads), demonstrating moderately elevated perfusion on DSC lc-rCBV, but slightly less conspicuous on DSC rCBF and VSASL rCBF. Note very high (red) perfusion signal due to vessels along the Sylvian fissures bilaterally on DSC.

**Figure 2 brainsci-14-00126-f002:**
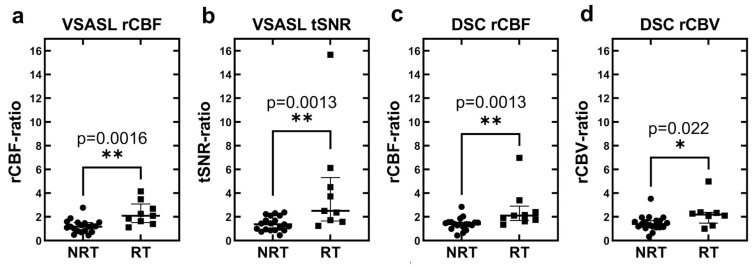
Dot–whisker plots of relative perfusion values of patients without and with tumor recurrence measured by (**a**) VSASL rCBF, (**b**) VSASL tSNR, (**c**) DSC-PWI rCBF, and (**d**) DSC-PWI lc-rCBV. In each plot, all points are included and represented in round or square dots from no recurrent tumor (NRT) and recurrent tumor (RT) groups, respectively. The middle bar represents the group median; the upper and lower bars represent the interquartile range. For all four perfusion parameters, there is significantly higher median values in RT compared to NRT cases. ** denotes significant level *p* < 0.005 and * denotes *p* < 0.05.

**Figure 3 brainsci-14-00126-f003:**
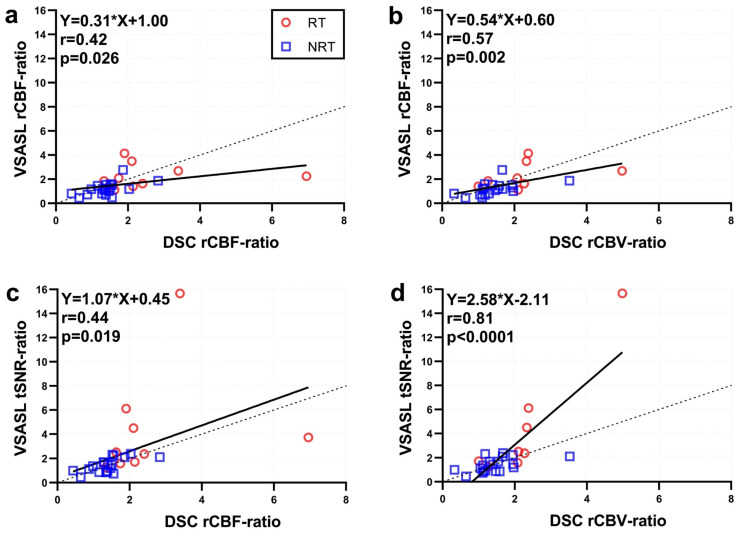
Concordance plots between the perfusion ratio values derived from VSASL and DSC. (**a**) VSASL rCBF vs. DSC rCBF, (**b**) VSASL rCBF vs DSC lc-rCBV, (**c**) VSASL tSNR vs. DSC rCBF, and (**d**) VSASL tSNR vs. DSC lc-rCBV. Tumor recurrence (RT) cases were coded in red, and no recurrent tumor (NRT) cases were denoted by blue. Solid black lines represent the linear regression curves with the fitted equations displayed at top left; dashed black line is the identity line.

**Figure 4 brainsci-14-00126-f004:**
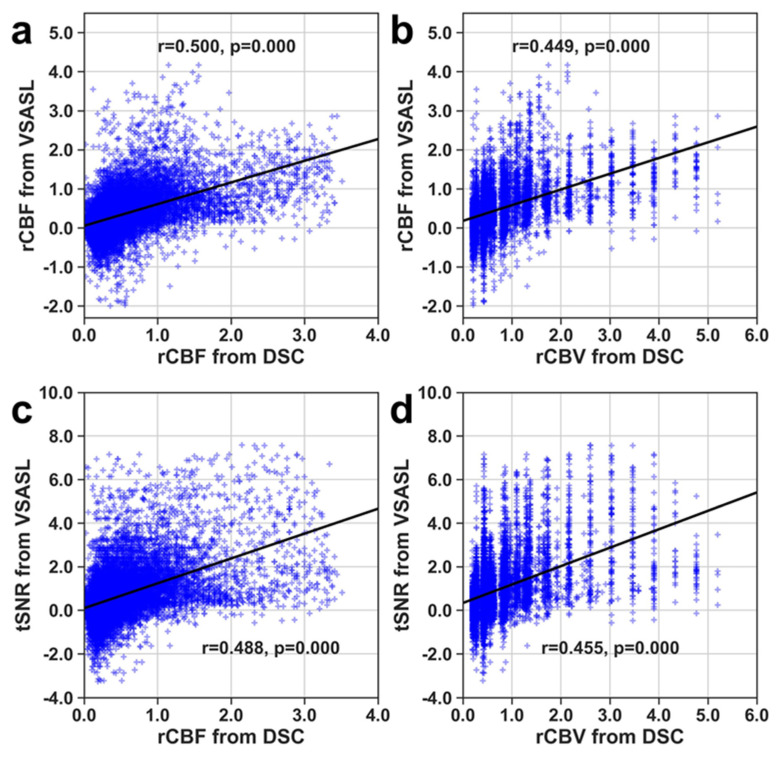
Spatial correlation of voxels within lesion ROIs of tumor cases by comparing VSASL and DSC. (**a**) VSASL rCBF vs DSC rCBF, (**b**) VSASL rCBF vs. DSC lc-rCBV, (**c**) VSASL tSNR vs DSC rCBF, and (**d**) VSASL tSNR vs. DSC lc-rCBV. Solid black lines represent the fitted linear regression curves.

## Data Availability

The data are not publicly available due to privacy or ethical restrictions. The data that support the findings of this study are available from the corresponding author upon reasonable request. The VSASL sequence and the in-house Matlab codes for post processing methodology used in this study are also available to share. The approval from the requesting researcher’s local ethics committee request is encouraged.

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
