# Peer review of "Velocity-Selective Arterial Spin Labeling Perfusion in Monitoring High Grade Gliomas Following Therapy: Clinical Feasibility at 1.5T and Comparison with Dynamic Susceptibility Contrast Perfusion"

_brainsci, 2024, doi:10.3390/brainsci14020126_

Round 1
Reviewer 1 Report
Comments and Suggestions for Authors
In this paper, applying routine clinical surveillance MR exams of 28 patients with high-grade gliomas at 1.5T, authors assessed the utility of velocity-selective ASL (VSASL) compared to the more commonly utilized DSC perfusion technique. A ROI was drawn to cover the largest dimension of the lesion with measures normalized against normal gray matter to yield rCBF and tSNR from VSASL, as well as rCBF and leakage-corrected relative CBV (lc-rCBV) from DSC. Lin’s concordance analyses showed moderate to excellent concordance between the two methods. These results suggest that VSASL is clinically feasible at 1.5T and has the potential to offer a non-invasive alternative to DSC perfusion in monitoring high-grade gliomas following therapy.
My main concerns are as follows:
1. In this paper, motion correction was applied using SPM12. How to perform motion correction is less clear.
2. All perfusion maps were co-registered with anatomic images. How to register the two modal images is not clear.
3. An ROI was drawn for each case on one single axial slice covering the largest dimension of tumor/lesion using ImageJ 1.52a to encompass areas of post-contrast T1 and FLAIR hyperintensity, while avoiding large vessels. Different ROI sizes have an impact on the calculated parameters. How to obtain a relatively better ROI?
4. The perfusion characteristics of glioma in different stages are different. Specific clinical staging of glioma samples will make the results more meaningful.
5. At present, 3T is widely used in clinical MRI scanning. Please discuss the situation of 3T briefly.
Reviewer 2 Report
Comments and Suggestions for Authors
The purpose of your study was the evaluation of the feasibility of VSASL incorporated into the routine clinical exam of gliomas at 1.5T and comparison of VSASL with DSC technique. While your introduction and methods part reads gratifying i would recommend to add some references that discuss the evolution of VSASL in more detail and much more up to date. Connected to this readers should be informed about a few questions that arise from using VSASL. How to mitigate artifacts of the velocity-selective profiles from B0/B1 field inhomogeneities ? Do eddy currents play a major role in VSASL quality? Can blood velocity variation during the pulses in case of arterial blood pulsation or blood vessel tortuosity, lead to signal loss for the flowing spins with fast-velocities? Please explain.
For the future use VSASL in clinical application the availability and usability of necessary software is a keypoint. Please make clear that all software you recommend to use is freely accessible. Would you provide SOPs for the workflow of analysis? Alternatively do you know sources of SOPs for the analysis pipeline and what will be the next step for routine application of your recommended method. Please discuss.
Round 2
Reviewer 1 Report
Comments and Suggestions for Authors
The authors have addressed all my concerns. I recommend acceptance for publication.
